# Study Protocol for a Hospital-to-Home Transitional Care for Older Adults Hospitalized with Chronic Obstructive Pulmonary Disease in South Korea: A Randomized Controlled Trial

**DOI:** 10.3390/ijerph20156507

**Published:** 2023-08-02

**Authors:** Heui-Sug Jo, Woo-Jin Kim, Yukyung Park, Yu-Seong Hwang, Seon-Sook Han, Yeon-Jeong Heo, Dahye Moon, Su-Kyoung Kim, Chang-Youl Lee

**Affiliations:** 1Department of Health Policy and Management, School of Medicine, Kangwon National University, 1 Kangwondaehak-gil, Chuncheon-si 24341, Republic of Korea; choice@kangwon.ac.kr (H.-S.J.);; 2Department of Internal Medicine, Kangwon National University, 1 Kangwondaehak-gil, Chuncheon-si 24341, Republic of Korea; pulmo2@kangwon.ac.kr; 3Department of Preventive Medicine, Kangwon National University Hospital, 156, Baengnyeong-ro, Chuncheon-si 24289, Republic of Korea; ykpark@knuh.or.kr; 4Department of Internal Medicine, Kangwon National University Hospital, 156, Baengnyeong-ro, Chuncheon-si 24289, Republic of Korea; ssunimd@kangwon.ac.kr (S.-S.H.); yonjong1954@knuh.or.kr (Y.-J.H.); ansekgo@knuh.or.kr (D.M.); 5Division of Pulmonary, Allergy and Critical Care Medicine, Department of Internal Medicine, Hallym University Chuncheon Sacred Heart Hospital, Hallym University College of Medicine, 77, Sakju-ro, Chuncheon-si 24253, Republic of Korea

**Keywords:** older adult, transitional care, chronic obstructive pulmonary disease

## Abstract

Chronic obstructive pulmonary disease (COPD) is a progressive respiratory condition characterized by persistent inflammation in the airways, resulting in narrowing and obstruction of the air passages. The development of COPD is primarily attributed to long-term exposure to irritants, such as cigarette smoke and environmental pollutants. Among individuals hospitalized for exacerbations of COPD, approximately one in five is readmitted within 30 days of discharge or encounters immediate post-discharge complications, highlighting a lack of adequate preparedness for self-management. To address this inadequate preparedness, transitional care services (TCS) have emerged as a promising approach. Therefore, this study primarily aims to present a detailed protocol for a multi-site, single-blind, randomized, controlled trial (RCT) aimed at enhancing self-management competency and overall quality of life for patients with COPD through the provision of TCS, facilitated by a proficient Clinical Research Coordinator. The RCT intervention commenced in September 2022 and is set to conclude in December 2024, with a total of 362 COPD patients anticipated to be enrolled in the study. The intervention program encompasses various components, including an initial assessment during hospitalization, comprehensive self-management education, facilitation of social welfare connections, post-discharge home visits, and regular telephone monitoring. Furthermore, follow-up evaluations are conducted at both one month and three months after discharge to assess the effectiveness of the intervention in terms of preventing re-hospitalization, reducing acute exacerbations, and enhancing disease awareness among participants. The results of this study are expected to provide a basis for the development of TCS fee payment policies for future health insurance.

## 1. Introduction

Chronic obstructive pulmonary disease (COPD) is characterized by airflow limitation, persistent respiratory symptoms caused by increased airway resistance, and lung parenchymal damage caused by exposure to noxious particles or gases [1]. Given its high morbidity and mortality worldwide, COPD is expected to be the third leading cause of death by 2030 [2]. The transition period, when patients are discharged from the hospital and sent home, is important because they have to manage the disease independently. According to previous studies, one out of five patients with COPD hospitalized for acute exacerbations was either re-hospitalized within 30 days of discharge [3,4] or experienced adverse effects immediately after discharge and was unprepared for self-management [5]. It is highly possible that patients are unable to cope with the disease owing to a lack of skills in using an inhaler [6] and not recognizing the symptoms of acute exacerbation [7]. Transitory care services (TCS) have emerged as an alternative solution to these problems. A TCS is a patient-centered treatment model designed to improve the quality of life and treatment of patients with chronic diseases and their families. In particular, it is an intervention method that ensures safe and effective treatment and continuity of patient management in the process of moving to a home, nursing hospital, or long-term care facility. TCS establish a discharge and transfer plan based on an in-depth evaluation of the patient from the time of hospitalization and share information about patient care among patients, their families, and healthcare workers [8].

Previous studies have reported that, by providing TCS to COPD patients, their readmission and hospitalization periods decrease [9], and their dyspnea [9], quality of life [10], and depression scores [11] improved. Specifically, in the study by Ko et al. [9], a respiratory nurse provided two hours of education on drug use, smoking cessation, and respiratory management to patients discharged from the hospital owing to exacerbation of COPD while a physical therapist performed pulmonary rehabilitation. In addition, when patients had questions about the disease, they could call a respiratory nurse for consultation. Consequently, the hospitalization period due to readmission, acute exacerbation, and the degree of dyspnea were reduced in the intervention group (IG). Similarly, in Xu’s study [10], patients with COPD who were admitted to the hospital because of acute exacerbation were provided self-management education. After discharge, health guidance was provided via phone or messaging. As a result, quality of life improved in the IG group. Finally, in Hegelund [11], advice on health care was provided over the phone for three months after discharge according to the severity of respiratory symptoms in COPD patients. Depression scores improved in the IG group.

Our ongoing study attempts to provide a comprehensive and evidence-based TCS intervention program by bundling intervention elements (Patient-tailored education, Community connection, Home [or Facility] visit, Telephone consultation, Multidisciplinary care plan meeting, etc.) that are effective in strengthening the self-management competency of patients with COPD after discharge. In contrast to previous studies, this study employs a Delphi survey using the UK’s “National Health Service (NHS) Outcomes Framework” [12] to reflect broad indicators including not only “effectiveness” indicators, such as re-hospitalization and acute exacerbation experience, but also indicators of drug intake in terms of “patient safety” and patient experience evaluation in terms of “patient-centeredness” into randomized, controlled trial (RCT) outcome indicators. Moreover, as the first TCS program conducted in South Korea, which was specifically designed for patients with COPD, this study develops COPD education materials tailored to the Korean context. These materials encompass essential aspects of self-management following hospital discharge, including inhaler usage and acute exacerbation management techniques and strategies. Furthermore, in recognizing the significance of post-discharge self-management, we introduce a management diary to empower patients to set personalized goals and enhance their adoption of health-promoting behaviors.

## 2. Study Design

This study is a randomized, controlled, and single-blinded trial consisting of an Intervention Group (IG) and Control Group (CG). It was approved by the institutional review boards of Kangwon National University Hospital (IRB number KNUH-B-2022-08-003-004) and Hallym University Chuncheon Sacred Heart Hospital (IRB number CHUNCHEON-2022-07-002-001). It is also registered with the Clinical Research Information Service (number KCT0007937). The patient, intervention, comparison, and outcome (PICO) model for this study has been set as follows.

(1)Patients: This study is a multi-center study, in which Kangwon National University Hospital and Hallym University Chuncheon Sacred Heart Hospital, both located in Chuncheon, participated with the aim of enrolling 362 patients (IG = 181, CG = 181) hospitalized with COPD at these two hospitals.(2)Intervention: For patients assigned to the IG, the clinical research coordinator (CRC) in charge visits their hospital rooms during hospitalization to conduct an initial evaluation, and the medical social welfare team visits them once to assess the need for and connect them to welfare services. Later, the CRC visits the hospital room twice and educates the patients on self-management using a booklet. In addition, when patients are discharged to a home or nursing facility, the CRC visits the home or nursing facility twice to educate and train them on self-management, and if they require additional interventions, their health conditions are checked through 3–4 follow-up phone calls (F/Us).(3)Comparison: Participants assigned to the CG receive general care upon discharge from the hospital, and similar to the IG, the CRC visits the ward during hospitalization and conducts an initial evaluation.(4)Outcomes: Readmission, emergency room visits, clinical respiratory indicators (COPD assessment test (CAT)), modified medical research council dyspnea scale (mMRC), acute exacerbation, etc.), patient-centered indicators (patient experience), etc., are measured. The participants are not made aware of whether they are included in the IG or CG, and this status is not disclosed until the end of the study to prevent contamination of the research results.

### 2.1. Setting

This study serves as a comprehensive protocol paper that outlines the implementation of TCS by CRCs for patients hospitalized with COPD. The primary aim of this study is to enhance patients’ compliance with self-management strategies following their discharge, ultimately leading to a reduction in re-hospitalization rates and an overall improvement in quality of life. This study of “evidence-based discharge management for high-risk discharged patients and evaluating the effects of the community connection program” is supported by the National Evidence-Based Healthcare Collaborating Agency. Having begun in September 2022, we intend to conduct this health management program until December 2024 following COPD patients’ discharges from Kangwon National University Hospital and Hallym University Chuncheon Sacred Heart Hospital in Chuncheon, South Korea.

### 2.2. Participants

Patients aged 18 years or older who were diagnosed with COPD and admitted to Kangwon National University Hospital and Hallym University Chuncheon Sacred Heart Hospital were selected as study participants at the beginning of the study.

### 2.3. Identification and Recruitment

During the screening for patient recruitment, the CRC identified the participants within 72 h after admission to the hospital by checking their diagnoses in the hospitals’ electronic medical records (EMR) systems. Participants with the diagnostic code for COPD (J44.0, J44.9) as the main diagnosis or sub-diagnosis and the result of a pulmonary function test showing Forced Expiratory Volume in one second(FEV1)/Forced Vital Capacity (FVC) < 0.70 on the EMR were considered to meet the clinical eligibility criteria. The CRC confirmed that patients were appropriate to be enrolled in a clinical study by their primary respiratory physician. The primary respiratory physician received a list of eligible patients from the CRC and was requested to indicate which patients should be included or excluded from the study. The exclusion criteria are: (1) patients receiving active treatment owing to a cancer diagnosis; (2) patients with mental and behavioral disorders who faced difficulty participating in the study (survey and education compliance); and (3) patients diagnosed with dementia. The study period procedure is summarized in a Consolidated Standards of Reporting Trials (CONSORT) diagram (Figure 1).

### 2.4. Enrollment and Randomization

The procedure for assigning the participants is as follows. A primary care physician explains the purpose of the study to patients with COPD admitted to the hospital, and oral consent is obtained. Subsequently, the research team proceeds with stratified randomization according to the most recent lung function values that could be confirmed in the EMRs. Random assignment is stratified into mild (FEV1 50% or more) and severe (FEV1 50% or less) using lung function values to ensure similar distribution of the clinical severity in the IG and CG. Among the mild and severe lung function values, the IG and CG are randomized. After the CRC explains the study in detail to the patients according to the allocation results, the primary respiratory physician obtains written informed consent from the patient. The CRC coordinates surveys and education schedules on health management (e.g., inhaler use, breathing management, symptom management) for patients in the IG. The CRC conducts a 40-min verbal survey of patients using a questionnaire and then inputs the response results into the Electronic Case Report Form (E-CRF) of the Internet-based Clinical Research and Trial Management System (iCReaT Ver.2), operated by the National Institute of Health for data quality management. E-CRF can efficiently manage the clinical research database design and management, data storage, backup, and security [13], and it can increase data accuracy, reduce clinical development costs, and efficiently manage processes [14]. For patients participating in the IG, the CRC provides educational materials in the form of a booklet and conducts education sessions twice during hospitalization and twice after discharge. In addition, a social worker visits the hospital room to assess the need for welfare services and connects the patient with them. Participants in the IG and CG receive gift certificates worth $15 if they complete the pre-survey and $7.50 each if they complete the post-survey and follow-up. The detailed process for the registration of the research participants is shown in Figure 2.

### 2.5. Intervention Group

#### 2.5.1. CRC Training and Support

CRCs are those who have graduated from a four-year nursing college, have a nursing degree, and have completed 40 h of a new clinical trial coordinator course to enhance their clinical trial-related work performance and understanding. “Regulations and applications related to clinical trials (2 h)”, “consent and management of test participants (including practice (4 h)”, “review and preparation of case reports (2 h)” and “monitoring and reporting adverse event (2 h)” are among the 20 topics covered in the training. When all the training is complete, CRCs take the exam and must score 60 points or higher. After completing the new training program, CRCs strengthen their expertise in clinical trials by completing an advanced course for CRCs (more than 24 h) and continuing education (more than 8 h) once per year. Specifically, the advanced course designed for CRC training, consisting of 14 h of an offline education program for one night and two days, encompasses various pertinent topics, such as “new drug development and pharmacokinetics/pharmacodynamics”, “bioequivalence testing”, “initial clinical trials”, “protocol development for major efficacy measures”, and “a comparison analysis of domestic and international regulations pertaining to clinical trials”. The online advanced courses include Good Clinical Practice (GCP) training (6 h) and a basic statistics in clinical trials (4 h) module. CRC continuing education includes “multicenter/multinational clinical trials and regulations for multinational clinical trials (1 h)”, “research fund calculation and insurance subscription for researcher-led clinical trials (1 h)”, and continuing courses for GCP training (4 h) [15].

CRCs receive training using the “Practitioner Competency Building for COPD Patient Health Management Program after Discharge” curriculum. The content of the curriculum consists of modules on educating patients with COPD, communication, and CRC coaching. The module for educating patients with COPD imparts knowledge and understanding of COPD and the transitional care model (3 h), the use of oxygen and inhalants (1 h), and home visits (1 h). Communication and coaching consist of active listening and verbal and nonverbal communication methods to strengthen patients’ motivation (2 h). Specifically, the education for “strengthening the capacity of healthcare program practitioners after COPD patients are discharged” is conducted by preventive medicine and pulmonologist doctors, nursing professors, and representatives of specialized communication education institutions. CRCs are taught the importance of discharged patient management, community connection, COPD symptoms and diagnosis, the Global Initiative for Chronic Obstructive Lung Disease guidelines [16], the international guideline for COPD management, and how to manage acute exacerbations. In addition, CRCs learn techniques for using inhalers, smoking cessation counseling, the importance of pulmonary rehabilitation, pursed-lip breathing, and abdominal breathing. 

#### 2.5.2. Initial In-Depth Evaluation and Establishment of Care Plans 

Care plan meetings are held once per week to establish and check care plans through interdisciplinary discussions among researchers and to improve the quality of interventions. Doctors, CRCs, clinical research associates, social workers, and researchers participate in the care plan meetings. Based on the initial survey results, the need for education on respiratory disease management, the need for education on inhaler use and drug intake, the need for smoking and living (ventilation method guidance, etc.), and management guidance are planned. Based on the established care plan, education is provided during visits to participants’ homes. In addition, at a meeting attended once per month by primary respiratory physicians, the health status of each participant is examined, and the part requiring connection with family medicine (a smoking cessation program) is discussed to prevent acute deterioration and re-hospitalization of COPD patients. 

#### 2.5.3. Patient-Tailored Education and Respiratory Rehabilitation

Patient-tailored education is conducted in four sessions. The first two sessions are conducted on two separate occasions by the CRC while the patient is hospitalized. The third and fourth sessions are conducted at the patient’s home or facility 48 h and 1 month after being discharged, respectively. The Montreal Chest Institute’s Living Well with COPD (LWWCOPD) [17] was modified and translated to suit the circumstances of this study and was used as educational material. The LWWCOPD is an evidence-based self-management program for patients, designed to allow them to set their own learning goals and learn content at their own pace. The detailed contents include 10 modules: (1) what is COPD?; (2) major symptoms of COPD; (3) causes of exacerbation of COPD; (4) use of an inhaler; (5) breathing management; (6) breathing techniques; (7) daily management; (8) health management; (9) symptom management; and (10) management diaries.

Regarding pulmonary rehabilitation, the CRC educates patients on and encourages them to use pulmonary rehabilitation exercise methods through educational brochures and videos so that patients can train themselves on inhaler use and breathing techniques in their daily lives.

#### 2.5.4. Community Connection

While patients in the IG are hospitalized, those with CRCs are given social welfare cooperation. Social workers visit the hospital room and evaluate mobility, household type, socioeconomic status, etc. Through consultation and cooperation with local medical institutions, community centers, and welfare institutions, CRCs link services, such as applications for long-term care or disability grade, residential environment improvement, daily life and care support, and economic support. 

#### 2.5.5. Home (or Facility) Visit 

When an IG participant is discharged from the hospital, the CRC visits that participant’s home or facility twice (48 h and 1 month after discharge) to check whether he or she is continuing to use the education received at the time of hospitalization and to assess he patient’s overall health status (vital signs, etc.). In addition, any factors that impeded respiratory health are examined by checking the living environment to determine whether ventilation, proper indoor temperature, and humidity are maintained. The risk factors for falls are also examined. For an efficient examination, the CRC provides a visit log that summarizes the main results of the initial evaluation when visiting, and the visit log is structured so that the items can be checked at the time of the visit by comparing them with the initial evaluation results. The items in the visit log are divided into three categories: (1) COPD status; (2) self-management checklist; and (3) education. First, the COPD status item is designed such that the current status can be answered on a scale from 0 (good condition) to 5 points (bad condition), compared to the items expressing the degree of cough, sputum, and chest tightness in the initial evaluation. Second, the checklist for self-management, awareness (recognition of disease name and disease state), smoking cessation (smoking status, need for anti-smoking education), inhaler use (whether inhaler is used, adherence to an inhaler), medication adherence, physical activity (use of assistive devices and exercise), life management (sleep, meals, anxiety, and depression), and vaccination status is examined. Third, the necessary education is provided to the target person using educational materials according to the plan established at the care plan meeting. 

#### 2.5.6. Telephone Consultation

One month after the patient is discharged from the hospital, the CRC calls once per week to check the patient’s health. If patients respond with “as usual” when asked about his or her subjective health status, then the CRC checks whether the patient is still using the education that he or she was provided on inhaler use, breathing techniques, diet, smoking, etc. If patient responds with “worse than usual”, then the CRC asks follow-up questions including, “Is it difficult to breathe?”, “Do you cough more often than usual?”, “Is your phlegm darker than usual?”, and “Does your throat feel swollen and hot?” If patient answers “yes” to two or more questions, the CRC recommends an outpatient visit and helps with the reservation. 

### 2.6. Control Group

Patients assigned to the CG undergo pre-evaluation upon admission to the hospital and receive health management information upon discharge. Information materials comprise content on smoking cessation, drug-taking respiratory rehabilitation, and inhalant management. A follow-up evaluation is conducted when the patient visits the hospital as an outpatient 1 month after discharge, and another follow-up evaluation is conducted 3 months after discharge. The contents are listed in Table 1.

### 2.7. Measures

A self-report questionnaire is administered three times: when the patient is admitted to the hospital and 1 and 3 months after being discharged. The contents of the questionnaire include readmission, number of acute exacerbations, mMRC, CAT, and Hospital Anxiety and Depression (Figure 3). The experience survey at discharge is evaluated only during the initial survey, and the Partners at Care Transitions Measure (PACT-M), which evaluates the quality and safety of care when patients transition from hospital to home, is measured only at 1 and 3 months after discharge. The CRC also reviews the medications that patients are taking and observes patients demonstrating the use of inhalers. The detailed indicators are presented in Appendix A.

### 2.8. Outcomes

The primary outcomes include hospitalization-related indicators, such as readmission or emergency room visits, and respiratory-related indicators, such as exacerbation history, mMRC [18], CAT [19], and Test of the Adherence to Inhalers [20]. The secondary outcomes include the Hospital Anxiety and Depression Scale [21], self-efficacy [22], and Transition Care Quality Assessment [23].

### 2.9. Data Management and Quality Assurance (QA)

The data collected through questionnaires, phone calls, and visit records are recorded in the E-CRF under the supervision of the CRC. To ensure data accuracy, the data manager rigorously reviews the registered information, identifying errors in data registration, inconsistencies with questionnaires, and serious adverse events in the process. On confirming the integrity of the data, the database is securely locked, preventing any unauthorized modifications or tampering with the data. When an important data error is detected within the research team’s domain, it is promptly reported to the research director for verification. Following confirmation, the data manager approves the necessary corrections to be made by unlocking the database. Additionally, in the unfortunate occurrence of a research participant’s demise during the intervention (referred to as “study off”) or the participant being transferred to another hospital, leading to discontinuation of the intervention (referred to as “dropout”), the relevant details, including the specific intervention performed and the reasons for cessation, are meticulously documented in the E-CRF. Subsequently, the data undergo the locking process once again, following a thorough examination by the data manager, thereby ensuring the preservation of data integrity, as previously mentioned.

The principal investigator establishes the protocols before the initiation of the clinical trials. The CRA then monitors the data and compares them with supporting data to ensure that the clinical trial process is performed accurately to check whether matters related to the collection of clinical trial data, preparation of records and documents, and reporting comply with Korean Good Clinical Practices. In addition, conducting an internal audit, non-compliance management, and researcher training ensures that the research is conducted according to the institutional policy, and the protocol is approved by designating a QA auditor.

### 2.10. Sample Size Calculation 

We calculated the clinical research resource targets using G-Power software, version 3.1.9.7 (Heinrich-Heine-Universität Düsseldorf, Düsseldorf, Germany). Based on previous studies, the target value was confirmed by the readmission rate within 1 month as the main outcome indicator. According to existing data on inpatients with respiratory diseases at Kangwon National University Hospital, patients with COPD had a 26.2% re-hospitalization rate within 1 month of discharge. Referring to this fact and assuming that the COPD patients’ readmission rate in the participating hospitals was 26.2%, the readmission rate of the project group decreased to 15.6%. This outcome was expected owing to applying the meta-analysis results to the readmission reduction effect of transitional care. When estimated, based on 80% power and a 0.05 alpha error using the Z test, the final number of participants was estimated to be 362 (IG: 181, CG: 181). 

### 2.11. Statistical Analyses

Intention-to-treat analyses are conducted using Stata software, version 13.0 (College Station, TX, USA).

## 3. Discussion

This study aims to describe a TCS randomized, clinical trial designed to improve the disease management capacity of patients with COPD after discharge. The strengths of our TCS intervention program for patients with COPD are as follows.

First, this study is meaningful, as it is the first RCT to provide a TCS for patients with COPD in South Korea. This study was designed by applying Naylor’s [24] TCS main components (educating/promoting self-management, engaging, promoting continuity, collaborating, etc.), and the TCS model is applied for the first time. In other words, CRCs are employed to continuously provide self-management education from the time at which the participants are hospitalized to even after they are discharged. Care plan meetings involving a multidisciplinary team are held regularly to identify the research participants’ medical, economic, and social needs and to connect them with personalized healthcare and welfare services according to where they go after being discharged (home, rehabilitation facility, nursing facility, etc.) [22]. Based on the results of this study, an economic feasibility evaluation will be conducted and used as data to develop health insurance reimbursement policies in South Korea.

Second, the Kangwon National University Hospital and Hallym University Chuncheon Sacred Heart Hospital, from which the study participants were recruited, are university and general hospitals (total number of beds: 1037) with more than 500 beds [25] and a bed occupancy rate of 92.5% [26]. The two hospitals have the advantage of securing the regional representativeness of the recruitment sample, as they accommodate most patients who need admission to a general hospital within the catchment area, which covers the entire population of Chuncheon-si (approximately 280,000).

Third, a scoping review and Delphi survey are conducted to select the outcome indicators to identify the effectiveness of the TCS intervention program. First, the scoping review is set up as “effectiveness”, “patient safety”, and “patient centeredness” items, utilizing the healthcare quality framework presented by the Organization for Economic Cooperation and Development’s “Health Care Quality and Outcomes Indicators” [27] and the UK’s “National Health Service Outcomes Framework” [12]. Subsequently, a Delphi survey targeting experts (29 people) in the fields of medical care, health care, nursing, and social welfare is conducted for detailed outcome indicators in each category to ensure the appropriateness and validity of the indicators. In particular, the patient experience assessment, which has high priority as a detailed patient-centered item, is applied as the outcome index of this study to confirm whether healthcare services, corresponding to the needs of the study participants, are provided during the hospitalization period. Using the PACT-M [21], the research participants’ readiness for discharge and their long-term self-care experience at home and in facilities after discharge are measured. 

Fourth, the educational materials used in this study include essential self-management information for patients with COPD on the use of an inhaler and coping with breathing difficulties or acute exacerbations after discharge. A management diary is created to set goals for smoking cessation, exercise, symptom management, etc., and manage the progress. Setting patients’ goals through a management diary is meaningful because it contributes to the successful rehabilitation of patients with COPD by improving their intrinsic motivation and autonomy for disease management and increasing the possibility of healthy behavior [28,29]. In the intervention education section, topics related to respiratory rehabilitation, respiratory muscle training, and upper and lower extremity muscle strengthening exercises are provided through videos, in addition to pursed-lip breathing and diaphragmatic breathing. The use of instruments or tools is minimized to allow participants to easily accomplish the tasks at home by themselves, and content that could be performed with familiarity in daily life is included. 

Fifth, patients with COPD who are about to be discharged often experience delays in being discharged from or readmitted to medical institutions owing to their fear of leading daily lives in the community with the disease, which has been identified as a cause of increased healthcare expenses [30]. This study aims to help patients with COPD to adapt as members of the community by comprehensively evaluating the mobility, household type, and economic status of hospitalized patients by a social worker and continuously monitoring services, such as long-term care or disability grade applications, and daily life support in cooperation with local medical institutions, community centers, and welfare institutions. 

Sixth, in previous studies, participants visited a hospital or healthcare institution more than once to participate in an intervention program after discharge, which consumed travel time and resources [11]. In this study, the CRC visits patients’ homes or facilities to provide education after discharge from the hospital, evaluates their health conditions through telephone consultations, and makes outpatient appointments in cases of deterioration, thereby minimizing the physical movement of the participants, which has been identified as the main cause of low participation rates and high participant dropout rates in previous studies [11]. In addition, there is an advantage in that participants can receive psychologically comfortable education from the CRC in charge, who builds rapport with the participants while providing education during hospitalization and directly visits the participants’ residences. Moreover, this study is meaningful in that it is easy to provide personalized education to the participants, as the CRC can directly check their living environments and contribute to the health promotion of participants living far from the hospital by promptly responding to their desire to maintain their health.

Upon completion of this study, the results are expected to be used as follows. First, by conducting an economic evaluation, it will be possible to provide evidence for the development of a health insurance reimbursement policy for TCS for COPD patients. Second, reducing acute exacerbation and preventable re-hospitalization of COPD patients is expected to contribute to reducing unnecessary medical expenses and enabling efficient expenditure of health insurance finances. In addition, by improving COPD patients’ self-management competency, it is expected to contribute to improving physical, mental, and social health, as well as quality of life, by reducing difficulties in daily life after discharge, thus improving independent living.

This study has the following limitations. First, it follows a single-blind design, as the coordinator provides education in the hospital room during the participants’ hospitalization, and the intervention plan for the participants is established at a multidisciplinary care plan meeting attended by the attending physician. However, efforts are being made to obtain informed consent from each IG and CG participant after the assignment to maintain single blindness. Second, owing to the administrative and financial limitations of the study, the monitoring period is limited to the short term (three months after discharge), making it difficult to examine long-term indicators, such as mortality and quality of life.

## 4. Conclusions

This study aims to prevent re-hospitalization, improve quality of life, and reduce healthcare costs by educating COPD patients on health self-management during hospitalization and after being discharged through out-of-hospital visits and telephone counseling sessions. This study intends to confirm the effectiveness of managing discharged COPD patients with the help of hospitals and communities, and it suggests more efficient management methods.

## Figures and Tables

**Figure 1 ijerph-20-06507-f001:**
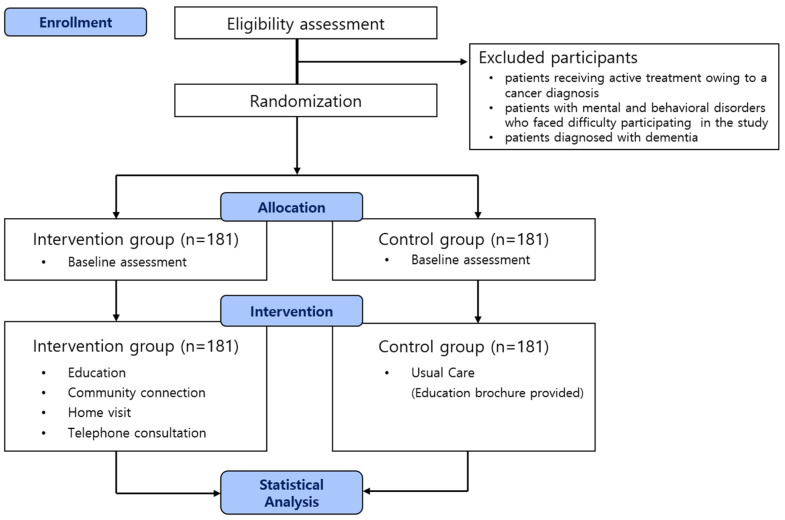
CONSORT diagram.

**Figure 2 ijerph-20-06507-f002:**
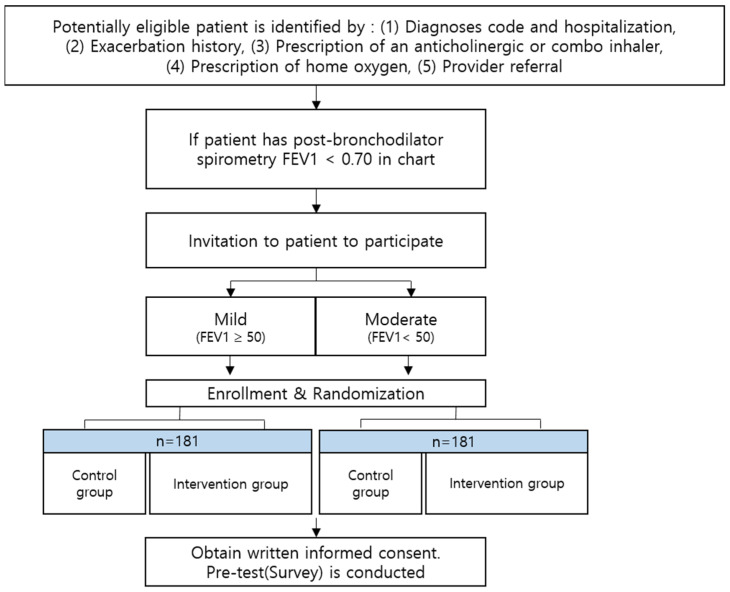
Study workflow from identification to enrollment.

**Figure 3 ijerph-20-06507-f003:**
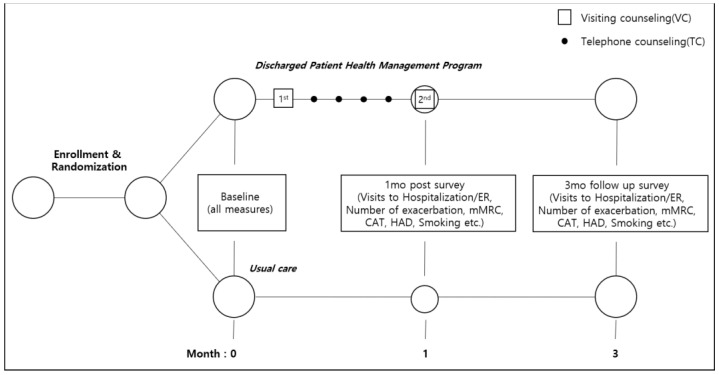
Participant timeline.

**Table 1 ijerph-20-06507-t001:** Summary of intervention.

	Study Period								
	Enrollment (t_1_)	Post-Allocation			Follow-Up				
		In the Hospital	Within 48 h before Discharge	Within 48 h of Discharge	1 Month (t_2_)				3 Months (t_3_)
Enrollment and assignment									
Informed consent	X								
Randomization of participants	X								
Baseline data collection	X								
Intervention (IG)									
Education		X	X	X	X				
Community connection		X							
Home visit				X	X				
Telephone consultation					X	X	X	X	
Usual Care (CG)									
Education brochure provided	X				X				X
Clinical-Data Collection									
mMRC, CAT, Test of the Adherence to Inhalers (TAI), hospitalization, emergency room visits, exacerbation history, patients’ experience, etc.	X				X				X

IG = Intervention group; CG = Control group.

## Data Availability

The data that support the findings of this study are available from the corresponding author upon reasonable request.

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
