# Peer review of "Study Protocol for a Hospital-to-Home Transitional Care for Older Adults Hospitalized with Chronic Obstructive Pulmonary Disease in South Korea: A Randomized Controlled Trial"

_ijerph, 2023, doi:10.3390/ijerph20156507_

Round 1
Reviewer 1 Report
There are issues with the structure of the manuscript.
The article lacks a results section, where the authors should present the results supported by statistical analyses as well as their interpretation. The conclusion section does not actually contain any conclusions from the study but is a repetition of its aim – what the study aimed, intended.
There are grammatical errors in English language (example from the abstract: “Patients with Chronic obstructive pulmonary disease(COPD) are suffer”). Please check the entire article for grammatical errors and correct them.
Figures add value and clarity to the article, but require editorial refinement in terms of punctuation - unnecessary spaces before colons (figure 1 and 2)."
There are grammatical errors in English language (example from the abstract: “Patients with Chronic obstructive pulmonary disease(COPD) are suffer”). Please check the entire article for grammatical errors and correct them.
Author Response
Thank you for this opportunity to revise and resubmit our manuscript. We also thank the reviewers for their helpful suggestions. The changes they offered have resulted in a much-improved manuscript. Please see our point-by-point responses to reviewer comments. Major text revisions are color (blue) coded for ease of review.

Reviewer 2 Report
Dear Authors,
It was very interesting to read your paper. Well done.
The only area I would suggest to improve is the results presentation.
The results should be clearly distinguished and shown emphatically. It's hard to find where they are shown. There is no results section.
Author Response

(The authors gave the same response as above.)

Reviewer 3 Report
Thank you for the opportunity to revise the manuscript "Study protocol for a hospital-to-home transitional care for older 2 adults hospitalized with Chronic Obstructive Pulmonary Disease in South Korea: A randomized controlled trial'. My comment are as follows:
1.Abstract: should not be in past tense
2. Introduction: The authors should try to explain what their study adds to the field
3.Methods:The objectives of the study should be clearly stated
4. The primary outcome should be clearly stated as well. Ideally should be only one that sample calculation was based on. In some cases two "co-primary outcomes" may be used. The authors should also list secondary outcomes.
5. Inclusion: for COPD studies usual inclusion criteria for age is >40yrs. Is there any reasoning for age >18yr cut off as this may increase the risk of misclassifying asthma patients as having COPD. Is smoking required criteria?
6. Please provide more data on IRB approval. As this is RCT written informed consent may be more appropriate than oral.
5.Data management plan and handling missing data should be described
6. The protocol should ideally describe in methods 1) design and development 2) implementation. Some of development is explained in the discussion (in scope review) but references were not provided.
7. The procedures should be described in more details ( when, how and by who) and what is expected impact. Materials used in program ( for examples checklist, structured interviews) should be provided (could be as an on line data supplement). Please also explain in more details training time and provide appropriate references for CRC training.
8. Discussion: The authors describe limitations but should also describe novelty and strenghts of their planned study
9. Statements in discussion should be cited with appropriate references
Abstract: should not be in past tense
Author Response

(The authors gave the same response as above.)

Round 2
Reviewer 1 Report
Dear Authors,
thank you for your explanations and the corrections made in the article as suggested. The article can be approved in it's current form.
Author Response
I'm appreciate your valuable time and opinion for the advancement of this paper.
